# Immunotherapy in Corticotroph and Lactotroph Aggressive Tumors and Carcinomas: Two Case Reports and a Review of the Literature

**DOI:** 10.3390/jpm10030088

**Published:** 2020-08-13

**Authors:** Camille Duhamel, Mirela Diana Ilie, Henri Salle, Adjoa Sika Nassouri, Stephan Gaillard, Elise Deluche, Richard Assaker, Laurent Mortier, Christine Cortet, Gérald Raverot

**Affiliations:** 1Endocrinology Department, Lille University Hospital, 59037 Lille, France; camille.duhamel5@gmail.com (C.D.); c.cortet@gmail.com (C.C.); 2Endocrinology Department, “C.I.Parhon” National Institute of Endocrinology, 011863 Bucharest, Romania; mireladiana.ilie@gmail.com; 3Neurosurgery Department, Limoges University Hospital, 87042 Limoges, France; Henri.SALLE@chu-limoges.fr; 4Endocrinology Department, Limoges University Hospital, 87042 Limoges, France; AdjoaSika.NASSOURI@chu-limoges.fr; 5Neurosurgery Department, Foch Hospital, 92150 Suresnes, France; s.gaillard@hopital-foch.org; 6Oncology Department, Limoges University Hospital, 87042 Limoges, France; Elise.DELUCHE@chu-limoges.fr; 7Neurosurgery Department, Lille University Hospital, 59037 Lille, France; r-assaker@chru-lille.fr; 8Dermatology Department, Lille University Hospital, 59037 Lille, France; l-mortier@chru-lille.fr; 9Endocrinology Department, Reference Center for Rare Pituitary Diseases HYPO, “Groupement Hospitalier Est” Hospices Civils de Lyon, 69677 Bron, France

**Keywords:** immune checkpoint inhibitors (ICIs), ipilimumab, nivolumab, prolactinoma, Cushing’s disease, aggressive pituitary tumor, aggressive PitNET, aggressive pituitary adenoma, pituitary carcinoma

## Abstract

Once temozolomide has failed, no other treatment is recommended for pituitary carcinomas and aggressive pituitary tumors. Recently, the use of immune checkpoint inhibitors (ICIs) has raised hope, but so far, only one corticotroph carcinoma and one aggressive corticotroph tumor treated with immunotherapies have been reported in the literature. Here, we present two cases, one corticotroph carcinoma and one aggressive prolactinoma (the first one reported in the literature) treated with ipilimumab (1 mg/kg) and nivolumab (3 mg/kg) every three weeks, followed by maintenance treatment with nivolumab (3 mg/kg every 2 weeks) in the case of the corticotroph carcinoma, and we compare them with the two previously reported cases. Patient #1 presented a biochemical partial response (plasma ACTH decreased from 13,813 to 841 pg/mL) and dissociated radiological response to the combined ipilimumab and nivolumab—the pituitary mass decreased from 37 × 32 × 41 to 29 × 23 × 42 mm, and the pre-existing liver metastases decreased in size (the largest one from 45 to 14 mm) or disappeared, while a new 11-mm liver metastasis appeared. The maintenance nivolumab (21 cycles) resulted in a stable disease for the initial liver metastases, and in progressive disease for the newly appeared metastasis (effectively treated with radiofrequency ablation) and the pituitary mass. Patient #2 presented radiological and biochemical progressive disease after two cycles of ICIs—the pituitary mass increased from 38 × 42 × 26 to 53 × 57 × 44 mm, and the prolactin levels increased from 4410 to 9840 ng/mL. In conclusion, ICIs represent a promising therapeutic option for aggressive pituitary tumors and carcinomas. The identification of subgroups of responders will be key.

## 1. Introduction

The vast majority of pituitary adenomas, recently renamed pituitary neuroendocrine tumors (PitNETs) [1,2], are benign and are easily treatable with already established therapeutic options—surgery; conventional medical treatments, including somatostatin receptor ligands and dopamine receptor type 2 agonists; and sometimes radiotherapy (RT) [3]. However, PitNETs may also prove to be aggressive and, very rarely, to metastasize, in the latter case being called pituitary carcinomas [1]. Both pituitary carcinomas and aggressive PitNETs lead to increased morbidity and mortality, and are very difficult to manage. The European Society of Endocrinology (ESE) guidelines on aggressive PitNETs and pituitary carcinomas recommend temozolomide, an oral alkylating agent, to be used after the failure of surgery, conventional medical treatments, and radiotherapy [3]. Unfortunately, in the ESE survey, the largest series on the use of temozolomide, this treatment led to a partial or complete radiological response in only 37% of cases, and to stable disease in 33% of cases. Moreover, after treatment ceased, progressive disease was noted in 25, 40, and 48% of patients initially responding to treatment with a complete response, partial response, and stable disease, respectively [4], and a second course of temozolomide proved to be effective only in rare cases [4,5,6]. Once temozolomide has failed, no other treatment option is formally recommended, because of a lack of sufficient evidence regarding efficacy in aggressive PitNETs, including carcinomas. Recently, the use of immune checkpoint inhibitors (ICIs) has raised hope, but so far only two cases of corticotroph tumors (one carcinoma treated with combined ipilimumab, an anti-cytotoxic T-lymphocyte-associated protein 4 (CTLA-4), and nivolumab, an anti-programmed cell death protein 1 (PD-1), which showed a partial response [7], and one aggressive corticotroph tumor treated with pembrolizumab alone, an anti-PD-1, which showed progressive disease [8]) are reported in the literature.

Here, we report two cases. First, we present the response of a functioning corticotroph carcinoma to a different dosing regimen of combined immunotherapy with ipilimumab and nivolumab than the one already reported in literature (chosen based on a better tolerance with comparable efficacy of this regimen in other cancers [9,10,11]). Second, we present the first reported case of an aggressive prolactinoma treated with immunotherapy.

## 2. Patients and Methods

Here, we present a case series describing the clinical management of one corticotroph carcinoma and one aggressive prolactinoma, with an emphasis on the effect of immunotherapy with ipilimumab (1 mg/kg every 3 weeks) and nivolumab (3 mg/kg every 3 weeks), followed by maintenance treatment with nivolumab alone in the case of the corticotroph carcinoma (3 mg/kg every 2 weeks). The cases were treated in two different French centers, Lille and Limoges, where they had been followed since 2001 and 2012, respectively. For both cases, the indication of immunotherapy was validated after a tumor board multidisciplinary discussion conducted at the national level (*HYPOCare* multidisciplinary reunion conducted by the French Reference Center for Rare Pituitary Diseases *HYPO*). Clinical, radiological, and pathological data were collected retrospectively from the patients’ medical records. All of the hormonal assays, immunohistochemistry, and methylation analyses were performed with commercially available kits. The study is in accordance with the Ethics Committee of Lille University Hospital and Limoges University Hospital. Informed consent was obtained from each patient after a full explanation of the purpose and nature of all of the procedures used, as well as for publication of the article along with accompanying images.

## 3. Patient #1

A 42-year-old female presented in 2001 with headaches, diabetes, hypertension, and obesity, and was diagnosed with Cushing’s disease based on an elevated urinary free cortisol (UFC) level of 212 µg/24 h (N 20-110), elevated morning plasma adrenocorticotropic hormone (ACTH) of 115 pg/mL (N < 46), and invasive pituitary tumor measuring 25 × 23 × 20 mm. The first transsphenoidal surgery (TSS), which was subtotal as a result of left cavernous sinus invasion, revealed a tumor with a Ki-67 index of 2%, absence of mitosis, and negative p53; therefore, it was classified as a grade 2a corticotroph tumor.

The regrowth of the tumor residue required multiple additional treatments between 2003 and 2017, as shown in Figure 1A, which shows the evolution of the plasma ACTH under the different treatments—fractionated RT (50 Gy in 28 fractions) in 2003; second TSS (rare mitosis) followed by Gamma knife radiosurgery (25 Gy) in 2007; 10 cycles of temozolomide 250–270 mg/day, 5 days every 4 to 5 weeks in 2009-2010 (with initial biochemical complete response and radiological partial response, but with relapse 3 years after treatment ceased; side effects: thrombocytopenia); three additional cycles of temozolomide 250 mg/day, 5 days every 5 weeks in 2013 (stopped due to radiological progressive disease and thrombocytopenia); third TSS (Ki-67 index 5%, 5 mitoses/10 high-power fields (HPFs), and positive p53–2%) in 2013; pasireotide 0.9 mg twice daily for 4 months in 2015 (ineffective); cabergoline 2 mg/week for 5 months in 2015-2016 (ineffective); hydroxyurea 1500 mg/day for 3 months in 2016–2017 (ineffective and causing pancytopenia); and fractionated RT (45 Gy in 25 fractions) in 2017.

In November 2018, the patient was admitted for asthenia, weight loss (10 kg), accentuation of melanoderma, right ptosis, and diplopia due to right third and sixth cranial nerve palsy. The plasma ACTH varied between 3957 and 11,191 pg/mL, compared with 398 pg/mL in May 2018. The pituitary computed tomography (CT) showed the global stability of the pituitary mass compared with the pituitary magnetic resonance imaging (MRI) performed in February 2018. The ^18^F-fluorodeoxyglucose (FDG) positron emission tomography (PET)/CT showed an intense uptake by the pituitary mass, the skull (corresponding to one lytic occipital lesion), and three hepatic lesions; there was also a suspected T2 vertebral lesion. The abdominopelvic CT showed five hepatic lesions, with maximal diameters of 31, 22, 12, 7, and 5 mm. A liver biopsy was performed and showed the presence of a corticotroph tumor metastasis (immunohistochemistry: Ki-67 index 10%, positive p53–7%, and absent programmed death ligand 1 (PD-L1) expression), confirming the diagnosis of pituitary carcinoma. The pituitary and thoraco-abdominopelvic CT repeated in February 2020 showed the progression of the pituitary mass (anteroposterior diameter: 37 mm vs. 32 mm) and of the two largest liver metastases (maximal diameters of 45, 30, 12, 7, and 6 mm). The evolution of the pituitary mass and of the liver metastases is presented in Figure 1B,C, respectively.

Immunotherapy with ipilimumab (1 mg/kg) and nivolumab (3 mg/kg) every 3 weeks was initiated in February 2019, and was administered for five cycles. A very rapid and significant decrease of plasma ACTH was observed after the first cycle, from 13,813 to 1036 pg/mL. The plasma ACTH continued to further decrease (nadir of 549 pg/mL after three cycles) and remained < 1000 pg/mL during the five cycles of combined immunotherapy. Regarding the radiological response, after three and four cycles, the pituitary mass decreased in size from 37 × 32 × 41 mm to 29 × 23 × 42 mm, while the five known liver metastases decreased in size from 45 to 14 mm, from 30 to 13 mm, from 12 to 5 mm, or were not visible anymore (for the smallest ones). The ^18^F-FDG-PET/CT showed a very positive metabolic response for the known metastases and the disappearance of the suspect T2 vertebral lesion, but the appearance of a new focal liver uptake was noted, corresponding to a new 11 mm lesion on the CT (described only afterwards on the CT). A concomitant improvement of melanoderma and right ptosis were observed, the diplopia disappeared, and the body weight stabilized. The patient did not report any side effects from the combined immunotherapy other than asthenia on the day of administration.

Since May 2019, the patient received maintenance treatment with nivolumab alone (3 mg/kg every 2 weeks). After the fourth cycle of nivolumab, the right ptosis worsened again, concomitantly with a progressive increase of plasma ACTH (1147 pg/mL, 1248 pg/mL, 1431 pg/mL, and 2094 pg/mL after the fourth, seventh, eleventh, and twelfth cycle in November 2019, respectively). The increase in the pituitary mass (from 29 × 23 × 42 to 29 × 25 × 45 mm in July 2019) was initially considered to be non-significant, however the pituitary CT performed in February 2020 confirmed the local progression (38 × 28 × 54 mm). Regarding the liver metastases, in November 2019, the thoraco-abdominopelvic CT showed the progression of the liver metastasis that became visible after the start of the immunotherapy (22 mm vs. 14 mm in July and 11 mm in May), whereas the other liver metastases were stable, measuring 12, 11, and 4 mm. Radiofrequency ablation of this liver metastasis was performed in January 2020, and 1-2 months later, the concurrent stigma of the radiofrequency ablation on the CT scan, a complete metabolic response of the hepatic lesion on the ^18^F-FDG-PET/CT, and a decrease of plasma ACTH were observed. The three other liver metastases remained stable, measuring 12, 8, and 4 mm. At the last follow-up in April 2020, the plasma ACTH was stable (1530 pg/mL) in comparison with March, but higher than in February (1250 pg/mL). The patient died four days later of an unknown cause. As potential side-effects of nivolumab, the patient reported asthenia, anorexia, and progressive weight loss (5 kg).

Of note, for the hypercortisolism, the patient also received ketoconazole and mitotane (before the biochemical remission induced by the 10 cycles of temozolomide), and after relapse, metyrapone 1000-3000 mg/day from August 2015 to December 2018 (side-effects: abdominal pain, hair loss, and hirsutism), and mitotane up to 2000 mg/day since February 2017 (side-effects: anorexia and vertigo).

## 4. Patient #2

A 60-year-old male presented in August 2012 with visual disturbances and bitemporal hemianopsia, and was diagnosed with a prolactinoma based on the presence of an invasive pituitary tumor measuring 36 × 31 × 26 mm, and a prolactin level of 5130 ng/mL (N 2.30-14.7). Figure 2A presents the evolution of the prolactin levels under the different treatments, while Figure 2B presents the evolution of the pituitary mass.

Cabergoline 0.5 mg × 3/week was started, and then increased to 0.5 mg daily from January 2013, enabling a decrease of prolactin to 1330 ng/mL, concurrent with a 50% decrease in tumor volume in March 2013. Cabergoline was further increased and maintained at 1 mg daily from June 2013. Despite an initial further decrease in the tumor volume and recovery of visual signs (nadir prolactin level of 787 ng/mL), the patient presented with the recurrence of the left temporal hemianopsia in December 2015. The pituitary MRI showed rapid tumor regrowth between July and December 2015 (from 7 × 19 × 17 mm to 30 × 25 × 25 mm). The TSS performed in January 2016 was subtotal as a result of left cavernous sinus invasion, and revealed a proliferative tumor (Ki-67 index of 10%, 5 mitosis/10 HPFs), and was therefore classified as a grade 2b prolactinoma. Adjuvant fractionated RT (50.4 Gy in 30 fractions) was performed in 2016. 

Following the first TSS, the patient was kept on 1 mg of cabergoline daily. At the end of 2018, the patient presented again with rapid tumor progression, with optic chiasm compression, for which a second TSS was performed in January 2019 (Ki-67 index 10–11%, 1 mitosis/10 HPFs). After surgery, cabergoline was further increased to a maximum of 10.5 mg/week and pasireotide 0.6–0.9 mg twice daily was administered for two months (ineffective and resulting in QT interval prolongation). The administration of temozolomide was decided, but due to the rapid tumor progression with a worsening of the visual field, decreasing visual acuity, and involvement of the left third cranial nerve, before starting temozolomide, a third TSS was performed in June 2019 (Ki-67 index 25%, >20 mitosis/10 HPFs). Neither the ^18^F-FDG PET/CT performed in June 2019 nor the medullary MRI performed in August 2019 found any metastasis.

The cabergoline was interrupted in July 2019, being considered ineffective given the rapid progression despite very high doses, and six cycles of 220–290 mg of temozolomide per day for 5 out of 28 days were administered starting at the end of July 2019. The tumor progressed under temozolomide, with worsening of the left cavernous sinus syndrome (diplopia, ptosis, mydriasis, and trigeminal neuralgia, for which methylprednisolone up to 80 mg daily was administered), a significant increase in the tumor size seen on the MRI performed in September 2019 (39 × 43 × 27 mm compared with 20 × 20 × 23 in June 2019), and a concurrent significant increase in prolactin levels. Then, 1 mg of cabergoline daily was reintroduced, enabling a decrease in prolactin levels, which, nevertheless, remained higher compared with the moment when the temozolomide was started. An improvement in the left cavernous sinus syndrome was also noted (possibly due to the methylprednisolone as well), but with the MRI performed in December 2019 not showing any regression in the pituitary mass (38 × 42 × 26 mm).

Given the inefficacy of temozolomide, the administration of immunotherapy with ipilimumab (1 mg/kg) and nivolumab (3 mg/kg) every three weeks was decided. The administration of methylprednisolone was progressively decreased, and stopped at the end of December 2019 in order to start the combined immunotherapy, while cabergoline was maintained at 1 mg daily. Unfortunately, the combined immunotherapy resulted in rapid progression, with the prolactin levels increasing from 4410 ng/mL before initiation to 9840 ng/mL after the first two cycles, and with the size of the pituitary tumor increasing from 38 × 42 × 26 mm in December 2019 to 53 × 57 × 44 mm in March 2020. Left ptosis also reoccurred. After the two cycles, the patient also presented severe side-effects (grade 3–4 diarrhea, and grade 1 nausea and vomiting), and therefore the combined immunotherapy was stopped. Starting at the end of March, 15 mg/kg of bevacizumab was administered every three weeks, with, so far, a stabilization of prolactin levels (~8830 ng/mL in April and May 2020), a stable disease shown in the MRI, and good tolerance (minimal epistaxis and grade 1 hypertension).

Of note, the somatostatin receptor type 5 (SST5) immunohistochemistry performed in January 2020 on the tumor block from January 2019 showed the absence of staining, which may partly explain the lack of an effect of pasireotide [12]. At the same time, as a potential marker of response to temozolomide, O6-methylguanine-DNA methyltransferase (MGMT) immunohistochemistry was performed, but was not interpretable, while the methylation analysis revealed a weak methylation of the MGMT promoter (average between 9 and 12%). On the tumor block from June 2019, genetic analysis was performed, as follows: (1) next-generation DNA-sequencing in January 2020—FoundationOne® companion diagnostic (CDx; Foundation Medicine, Inc., Cambridge, MA, USA)—revealing the absence of alterations with a known clinical and/or therapeutic impact (absence of microsatellite instability, low mutational burden (1 mutation/Mb), amplification of the *HGF* gene, deletion of the *CDKN2A/B* gene, mutation (p.Ser514*) of the *BCORL1* gene, mutation (p.Asn2fs*52) of the *FLCN* gene, and mutation (p.Lys666Glu) of the *SF3B1* gene), and (2) next-generation RNA-sequencing in March 2020—Archer^®^ FusionPlex^®^ CTL Panel (ArcherDX, Inc., Boulder, CO, USA)—revealing the absence of mutations or of fusion transcripts.

## 5. Discussion

ICIs block the inhibitory signals of T lymphocyte function and/or activation, and, more precisely, they block CTLA-4 or PD-1, which are found on T lymphocytes or its ligand, programmed death ligand 1 (PD-L1), which is found on antigen-presenting cells and on tumor cells. These inhibitory signals would otherwise enable tumors to evade immune response. By blocking them, ICIs enhance T lymphocyte function and reactivate antitumor immune responses [13,14,15]. 

Recent studies on the tumor microenvironment of PitNETs [16] have demonstrated in these tumors, and especially in functioning PitNETs, the presence of tumor-infiltrating lymphocytes [13,17,18] and of PD-L1, a potential biomarker of response to ICIs [13,17]. These findings have raised the hope that ICIs might be effective in these tumors. ICIs might be especially promising for patients who previously received conventional chemotherapies, such as temozolomide, because the administration of conventional chemotherapies can induce somatic hypermutations that will render ICIs more effective [7]. A recent in silico analysis of the immune tumor microenvironment of PitNETs also revealed that functioning corticotroph tumors had higher CD8+ T lymphocyte infiltration than somatotroph, lactotroph, thyreotroph, and non-functioning PitNETs, suggesting that functioning corticotroph tumors may be more amenable to ICIs than other PitNET subtypes [19].

So far, only two cases of corticotroph tumors have been treated with ICIs [7,8]. Table 1 summarizes their findings, together with the findings from our two cases. The two cases that showed a partial response were both carcinomas that had been previously treated with multiple conventional chemotherapies (including multiple courses), in comparison with the two cases that resulted in progressive disease, which were both aggressive pituitary tumors that had been previously treated with only one course of temozolomide. Although we did not perform genetic analyses before the start of the ICIs (and neither did the authors of the corticotroph tumor that showed progressive disease), a higher mutational burden would be expected in the two responders, and the liver metastasis of the responsive corticotroph carcinoma reported in the literature did indeed show somatic hypermutations [7]. However, it is worth mentioning that the aggressive corticotroph tumor that showed the progressive disease had a mismatch-repair deficiency (as did the liver metastasis from the other reported case), which, in other cancers, is associated with responsiveness to anti-PD-1 therapy [8]. Other than a less important mutational burden, additional possible reasons for ICI inefficacy in this case consists of using monotherapy instead of combined ICIs (less effective) and of the presence of high levels of cortisol during ICI administration, given that glucocorticoids have immunomodulating effects (the hypercortisolism was controlled in the other two cases). Moreover, this tumor did not express PD-L1 in the immunohistochemistry, but the predictive value of the PD-L1 expression still needs to be proven in pituitary tumors, especially given the fact that its predictive value has not been consistent across studies in other cancers [8]. Importantly, the liver metastasis that was biopsied in patient #1 did not express PD-L1 either, but proved to have an excellent response to ICIs, as did the liver metastasis from the other corticotroph carcinoma, which also only had <1% positive PD-L1 staining [7]. Therefore, the lack of PD-L1 expression should not preclude the indication of immunotherapy. Interestingly, both in our case and in the case of the corticotroph carcinoma previously reported [7], the liver metastases showed a greater decrease in size compared with the pituitary masses, which may be at least partly explained by a different composition of the tumor microenvironment between the liver and the pituitary (as two different anatomic sites), but also between primary and secondary lesions in general. In addition, given the rapid and significant biochemical response seen in the two responders (an ~10-fold decrease in ACTH levels within 1 week [7], and after the first cycle in patient #1), the biochemical response might prove to be an early and easy-to-perform marker of response to ICIs.

As limited experience is available on the use of ICIs in pituitary tumors, we think that response to treatment should be evaluated on a case-by-case basis. Given the regression of the pituitary mass and of all five pre-existing liver metastases, we consider patient #1 to be a responder to combined ipilimumab and nivolumab, instead of classifying the response as progressive disease based on the new liver metastasis that appeared after the four cycles of combined immunotherapy, as we could have done based on the iRECIST guidelines [20]. Although this last metastasis continued to grow on maintenance nivolumab and was finally treated by radiofrequency ablation, and the pituitary mass slowly progressed, nivolumab alone continued to be effective on the rest of the metastases. Therefore, in the case of a dissociate response (which might reflect a distinct evolution of the genetic/epigenetic landscape or of the tumor microenvironment), in which only one or very few of the lesions progress, the use of complementary local therapies such as radiofrequency ablation or surgery might be tried in combination with continued immunotherapy, instead of considering the immunotherapy ineffective and ceasing it. Additionally, the reintroduction of double immunotherapy might also prove useful in the future.

Moreover, besides a complete/partial response, stable disease, and progressive disease, one should be aware of two additional patterns of tumor response when using immunotherapy, namely: pseudoprogression (i.e., apparent increase in the tumor burden initially, followed by delayed tumor shrinkage, which could lead to the premature cessation of effective immunotherapy) [20,21] and hyperprogressive disease (i.e., accelerated tumor progression following the introduction of ICIs) [22,23]. Hyperprogressive disease was shown to be associated with older age—in a study including 131 patients treated with anti-PD-1/PD-L1 therapy, 7 out of 36 patients ≥65 years old (19%) were classified as having hyperprogressive disease, compared with 5 out of the 95 patients ≤64 years old (5%), *p* = 0.01, of which only one patient was <55 years old [22]. Patient #2 from the current study, who was 68 years old when immunotherapy was introduced, presented with rapid progression following ICIs initiation, but it is difficult to say whether it was the cessation of temozolomide, the natural history of the disease, or the introduction of ICIs that led to this evolution.

Regarding the side-effects of ICIs, combined immunotherapy, besides being more effective, also increases treatment-related adverse events [17]. In the four pituitary cases reported so far, the most frequently affected was the gastrointestinal tract, with patient #2 from the current study being the only one experiencing severe treatment-related adverse events. Regarding ICIs-related endocrinopathies, both of our patients had already substituted central hypothyroidism at the time immunotherapy started (for which the replacement therapy was maintained) and had type 2 diabetes mellitus (which remained well controlled during immunotherapy). No new endocrinopathies were diagnosed during immunotherapy.

## 6. Conclusions

So far, there is not enough data to enable any formal conclusion on the efficacy of ICIs in pituitary carcinomas and aggressive PitNETs, or on which subgroups of patients will prove to respond, but ICIs appear to be a promising therapeutic option. At the moment, there are two clinical trials running on combined ipilimumab and nivolumab, namely: NCT04042753, a clinical trial dedicated to pituitary carcinomas and aggressive PitNETs; and NCT02834013, a basket trial that also accepts pituitary carcinomas. Hopefully, their results will provide valuable information on the efficacy of ICIs in these tumors. At the same time, potential leads for the improvement of ICI efficacy emerging from studies on other cancers, such as the combination of ICIs with drugs targeting angiogenesis [24,25] or with radiotherapy [26,27], will potentially be transferable to pituitary carcinomas and aggressive pituitary tumors as well and will result in better outcomes for the patients treated with such therapies.

## Figures and Tables

**Figure 1 jpm-10-00088-f001:**
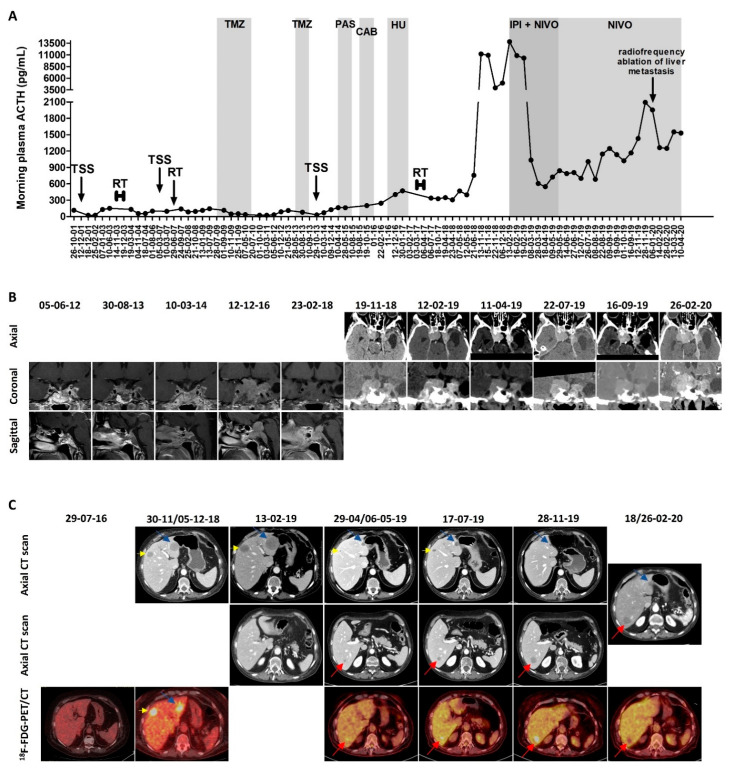
18-year follow-up of a corticotroph carcinoma (patient #1), from diagnosis to the present day. (**A**) Evolution of morning plasma adrenocorticotropic hormone (ACTH) levels under the different treatments. Every dot represents an individual measurement. (**B**) Radiological evolution of the pituitary mass as seen on the magnetic resonance imaging until February 2018 (contrast-enhanced T1-weighted images) and on the pituitary computed tomography from November 2018. (**C**) Radiological and metabolic evolution of the liver metastases. An ^18^F-fluorodeoxyglucose (FDG)-PET/CT performed on 29 July 2016 shows the absence of liver metastases. The blue and yellow arrows indicate the two largest liver metastases that appeared in November 2018. The metastasis indicated by the yellow arrow is not visible at every time point in this figure, but was still present, measuring 30 mm in February 2019, 13 mm in April 2019, 11 mm in November 2019, and 8 mm in February 2020. The red arrow indicates the liver metastasis that appeared after four cycles of combined immunotherapy. Abbreviations: TSS—transsphenoidal surgery; RT—radiotherapy; TMZ—temozolomide; PAS—pasireotide; CAB—cabergoline; HU—hydroxyurea; IPI—ipilimumab; NIVO—nivolumab; CT—computed tomography; PET—positron emission tomography.

**Figure 2 jpm-10-00088-f002:**
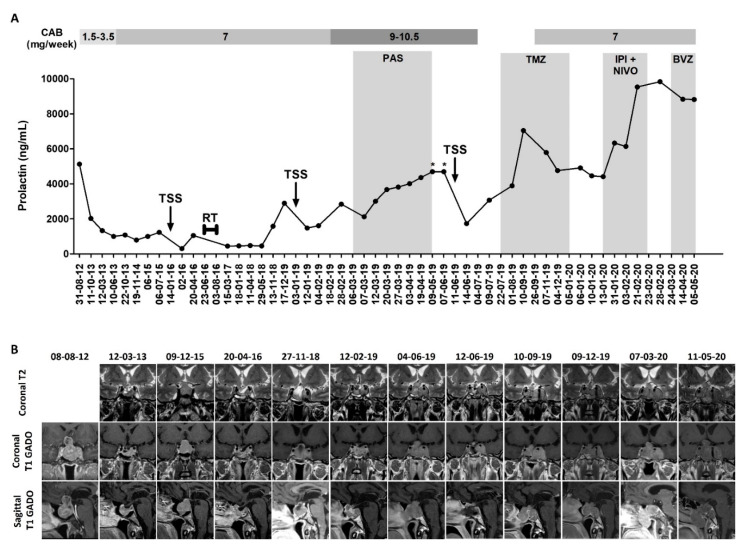
Eight-year follow-up of an aggressive prolactinoma (patient #2), from diagnosis to the present day. (**A**) Evolution of the prolactin levels under the different treatments. Every dot represents an individual measurement. * ≥4700 ng/mL (the upper limit of the respective prolactin assay). (**B**) Radiological evolution of the pituitary mass as seen on the magnetic resonance imaging. Abbreviations: CAB—cabergoline; TSS—transsphenoidal surgery; RT—radiotherapy; PAS—pasireotide; TMZ—temozolomide; IPI—ipilimumab; NIVO—nivolumab; BVZ—bevacizumab; T2—T2-weighted image; T1 GADO—contrast-enhanced T1-weighted image.

**Table 1 jpm-10-00088-t001:** Cases of pituitary carcinomas and aggressive pituitary neuroendocrine tumors treated with immunotherapy.

Ref.	Sex and Age ^1^	Tumor Type	Previous Tumor-Directed Treatments	Immunotherapy		
				Posology	Response	Side Effects
[7]	**F, 41**	**Corticotroph carcinoma**	NS × 2, RT, NS × 2, pasireotide, cabergoline, TMZ + capecitabine × 2 (4 and 2 cycles), etoposide + carboplatin (2 cycles), RT	IPI (anti-CTLA-4) 3 mg/kg + NIVO (anti-PD-1) 1 mg/kg every 3 weeks (5 cycles)	R: partial response: 59% ↘ in primary tumor volume and 92% ↘ in main liver metastasis volume B: partial response: plasma ACTH ↘ from 45,550 to 66 pg/mL	Fever (40 ℃), mild transaminitis, possibly hypophysitis
				Maintenance NIVO	R: “continues to respond” B: stable disease: plasma ACTH of 59 pg/mL at the 6-month follow-up	No additional immunologic side effects
[8]	M, 66	Aggressive corticotroph tumor	NS × 2, RT, TMZ for 2 years, pasireotide, NS	Pembrolizumab (anti-PD-1) 200 mg flat dose (4 cycles)	R: progressive disease, i.e., ≥ 20% ↗ in the sum of diameters B: progressive disease: plasma ACTH ↗ from 269 to 544 ng/L	None
	F, 60	Corticotroph carcinoma	NS, RT, NS, RT, TMZ × 2 (10 and 3 cycles), NS, pasireotide, cabergoline, hydroxyurea, RT	IPI 1 mg/kg + NIVO 3 mg/kg every 3 weeks (5 cycles)	R: partial response: primary tumor ↘ from 37 × 32 × 41 to 29 × 23 × 42 mm; the 5 liver metastases ↘ (the 2 largest ones from 45 to 14 mm and from 30 to 13 mm) or disappeared + a new 11 mm liver metastasis appeared B: partial response: plasma ACTH ↘ from 13,813 to 841 pg/mL	Asthenia on the day of administration
				Maintenance NIVO 3 mg/kg every 2 weeks (21 cycles)	R: stable disease for the initial 5 liver metastases + progressive disease of the newly appeared metastasis and of the pituitary mass B: progressive disease: plasma ACTH ↗ from 841 to 1954 pg/mL after 14 cycles	Possibly asthenia, anorexia, and progressive weight loss
	M, 68	Aggressive prolactinoma	Cabergoline (ongoing), NS, RT, NS, pasireotide, NS, TMZ (6 cycles)	IPI 1 mg/kg + NIVO 3 mg/kg every 3 weeks (2 cycles) + cabergoline 1 mg/day	R: progressive disease: ↗ from 38 × 42 × 26 to 53 × 57 × 44 mm B: progressive disease: prolactin levels ↗ from 4410 to 9840 ng/mL	Grade 3-4 diarrhea, grade 1 nausea and vomiting

^1^ The age of the patient when immunotherapy first started. Abbreviations: Ref.—reference; F—female; M—male; NS—neurosurgery; RT—radiotherapy for the primary tumor; TMZ—temozolomide; IPI—ipilimumab; CTLA-4—cytotoxic T-lymphocyte-associated protein 4; NIVO—nivolumab; PD-1—programmed cell death protein 1; R—radiological; B—biochemical; ACTH—adrenocorticotropic hormone.

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
