# Peer review of "Immunotherapy in Corticotroph and Lactotroph Aggressive Tumors and Carcinomas: Two Case Reports and a Review of the Literature"

_jpm, 2020, doi:10.3390/jpm10030088_

Round 1
Reviewer 1 Report
The authors reported two cases of aggressive PitNETs managed by immune checkpoint inhibitors. This is very interesting report, and a topic worth reading for many readers. The manner in which the authors presented the cases is adequate, which promotes the reader's understanding.
Minor comments)
Although the side effects of ICI administration were not reported in the text, ICI is known to cause various endocrinopathy. The manuscript will be improved further if the authors explain whether these patients have experienced any types of endocrinopathy, such as thyroid dysfunction or glucose fluctuation.
Reviewer 2 Report
The manuscript by Duhamel et al presents two case studies in which checkpoint inhibitors were used to treat pituitary tumors in patients following unsuccessful treatment by several first line chemotherapies. While the data are important for potentially improving treatment related decision regarding these rare types of cancers, the manuscript is not very well written, and contains several grammatical and formatting errors. My reservations and comments to the manuscript are provided below:
Command of the English language is not particularly evident as many sentences contain run-ons, or do not flow well within the paragraph. Sentences are repeated (Line 27-29 and Line 65-68) which is redundant. It would be useful to have it extensively reviewed and corrected grammatically prior to resubmission.
The title does not explain the manuscript well. A more detailed title would be helpful.
Line 28: add “treated with immunotherapies” to sentence. This makes it sound like only 2 have ever been reported.
Some sentenced could be written with more clarity. For example, Line 60-62 could be re-written “Moreover, after treatment was ceased, 25, 40, and 48% demonstrated disease progression for patients initially responding with a complete response, partial response, and stable disease, respectively.”
Line 64: lack of evidence of what?
Line 70-73: Rewrite for clarity (2 sentences would be easier to read and understand). Line 70: change "that" to "than."
Line 88: add “a” before “full”
Line 96: “therefore, it was classified as a stage 2a coritcotroph cancer.”
Figure legend 1 should be aligned on the left axis, not centered.
Line 101: 18F-FDG needs to be spelled out prior to its abbreviation.
Figure legend 2 should be aligned on left axis. An asterisk is used to denote levels above the detection limit of the assay; however, asterisks only appear on two dates (after PAS but before TSS), despite the presence of several point in which prolactin is over 4,700 ng/mL (see 08/19 – 05/20).
Line 205: “did not find” to “found”
Did either case show an influx of immunomodulatory cell phenotypes (effector T cells, macrophages, etc)? CD4/CD8 T cells present in the microenvironment have been attributed to response to ICI. Further, the immune landscape between liver and pituitary may be significantly different as does the pharmacokinetics of monoclonal antibodies at each tumor site.
References should start on new page
Author Response
Please see the attachment.

This manuscript is a resubmission of an earlier submission. The following is a list of the peer review reports and author responses from that submission.